# Enhanced quantum state transfer by circumventing quantum chaotic behavior

Liang Xiang [1,2,7], Jiachen Chen [1,2,7], Zitian Zhu [1,2,7], Zixuan Song [1,2], Zehang Bao [1,2], Xuhao Zhu [1,2], Feitong Jin [1,2], Ke Wang [1,2], Shibo Xu [1,2], Yiren Zou [1,2], Hekang Li [1,2], Zhen Wang [1,2], Chao Song [1,2], Alexander Yue [3], Justine Partridge [3], Qiujiang Guo [1,2] ✉, Rubem Mondaini [4,5,6] ✉, H. Wang [1,2] & Richard T. Scalettar [3] ✉

The ability to realize high-fidelity quantum communication is one of the many facets required to build generic quantum computing devices. In addition to quantum processing, sensing, and storage, transferring the resulting quantum states demands a careful design that finds no parallel in classical communication. Existing experimental demonstrations of quantum information transfer in solid-state quantum systems are largely confined to small chains with few qubits, often relying upon non-generic schemes. Here, by using a superconducting quantum circuit featuring thirty-six tunable qubits, accompanied by general optimization procedures deeply rooted in overcoming quantum chaotic behavior, we demonstrate a scalable protocol for transferring few-particle quantum states in a two-dimensional quantum network. These include single-qubit excitation, two-qubit entangled states, and two excitations for which many-body effects are present. Our approach, combined with the quantum circuit's versatility, paves the way to short-distance quantum communication for connecting distributed quantum processors or registers, even if hampered by inherent imperfections in actual quantum devices.

Among the many desired features of future large-scale quantum computation, the manipulation and transmission of quantum states without destroying their fragile coherence stand out as of primal importance. Originally, the transport of quantum information has been theoretically proposed[1] and experimentally demonstrated[2,3] by using entangled photons to mediate the information transfer between atom clouds over long distances, allowing quantum teleportation of states[4,5] and the implementation of quantum key-distribution[6,7], a fundamental step towards the realization of long-distance quantum secure communication[8]. Recently, the growing system sizes of quantum computing platforms[9–13] make it of paramount importance to realize quantum communication between different parts of a single device (or short-range quantum networks[14]), particularly for solid-state architectures with local interactions[11,15].

Considering short-distance quantum communication in solid-state devices, implementations in small chains[16–25] have primarily led the way. Particle transport in two-dimensional (2D) networks of superconducting qubits are further explored recently[26–28]. While the digital scheme of sequential SWAP gates provides a platform-independent way, the accumulation of minor two-qubit gate errors

[1]Zhejiang Key Laboratory of Micro-nano Quantum Chips and Quantum Control, School of Physics, Zhejiang University, Hangzhou 310027, China. [2]ZJU-Hangzhou Global Scientific and Technological Innovation Center, Zhejiang University, Hangzhou 311215, China. [3]Department of Physics and Astronomy, University of California, Davis, CA 95616, USA. [4]Beijing Computational Science Research Center, Beijing 100193, China. [5]Department of Physics, University of Houston, Houston, TX 77004, USA. [6]Present address: Texas Center for Superconductivity, University of Houston, Houston, TX 77204, USA. [7]These authors contributed equally: Liang Xiang, Jiachen Chen, Zitian Zhu. ✉e-mail: qguo@zju.edu.cn; rmondaini@uh.edu; scalettar@physics.ucdavis.edu

can ultimately hinder an efficient quantum state transfer (QST)[29]. An alternative approach, which avoids complex dynamical control of inter-qubit operations, is to use pre-engineered couplings that, in quantum circuits governed by a static Hamiltonian, achieve high-fidelity transfer of quantum information[30] (Fig. 1a).

Theoretical demonstration of this approach has been put forward in the case of an $N$-site one-dimensional (1D) $XY$-model quantum spin chain[31–33]:

$$\hat{H} = \sum_{\langle m,n \rangle}^{N} J_{m,n} [\hat{\sigma}_m^+ \hat{\sigma}_n^- + \hat{\sigma}_m^- \hat{\sigma}_n^+], \tag{1}$$

where $\hat{\sigma}_m^+$ ($\hat{\sigma}_m^-$) is the raising (lowering) operator for qubit $Q_m$, and the nearest-neighbor (NN) coupling between a pair of qubits is given by $J_{m,n}$. The key observation is that provided the couplings are chosen to satisfy $J_{n,n+1} = J\sqrt{n(N-n)}$ for $n = 1, ..., N-1$, the eigenvalues of Hamiltonian (1) in the single-excitation subspace are equal-spaced. It is equivalent to that of a large $(N-1)/2$-spin $\vec{S}$ under a homogeneous magnetic field, i.e., $\hat{H}/J = \hat{S}_+ + \hat{S}_- = 2\hat{S}_x$, where $\hat{S}_+$ ($\hat{S}_-$) is the raising (lowering) operator of the large spin. Such a 1D perfect QST scheme has been previously realized in small-scale superconducting circuits[17] and photonic qubits in coupled waveguides[16].

Generalization to higher dimensions is readily obtained in theory[34]. For example, in a bipartite lattice in $D$ dimensions, the constraints in the inter-qubit NN couplings satisfy a similar expression: $J_{n,n+1}^{(d)} = J\sqrt{n(N_d - n)}$ for $n = 1, ..., N_d - 1$ and $d = 1, ..., D$. The corresponding mapped large-spin Hamiltonian, $\hat{H}_D/J = 2(\hat{S}_{1,x} + \hat{S}_{2,x} + ... + \hat{S}_{D,x})$, describes a collection of $D$ large $(N_d - 1)/2$ spins, each independently precessing around its $x$-axis at the same rate, thereby guaranteeing perfect QST at time $tJ = \pi/2$ (Fig. 1d for

$D = 2$). In practice, however, even considering perfectly isolated systems without decoherence, parasitic cross-couplings[35], and device defects[11] can naturally occur and, as a result, hamper the perfect QST. The former introduces a connection between qubits across a plaquette via an unwanted coupling $J_{m,m'}^{\times}$ (Fig. 1b). After performing the mapping to the large-spin Hamiltonian,

$$\hat{H}_{\text{tot}} = 2J(\hat{S}_{1,x} + \hat{S}_{2,x}) + 4J^{\times} \hat{S}_{1,x}\hat{S}_{2,x}, \tag{2}$$

such extra terms result in a $J^{\times}$-mediated spin-spin interaction that spoils the standard predictions[31,32] of perfect QST (Fig. 1e). Here we assume for simplicity a single energy scale $J^{\times}$ that governs this term (see Methods). The latter, manifested as a defective coupler in our device (gray bond with a cross marker in Fig. 1b), similarly breaks the requirements of NN couplings for a perfect QST.

To overcome these limitations and efficiently transfer few-particle states in rather generic 2D networks, we utilize a Monte Carlo annealing process to find optimal NN coupling parameters that allow for a good QST under experimental limitations in tunability and extra constraints arising from imperfections (see Methods). With the optimized parameters, we realize efficient few-particle transfers in an imperfect 2D superconducting qubit network (Fig. 1b, c) by engineering NN couplings and unveiling the physical insights behind. Starting from a small $1 \times 6$ 1D chain and a $3 \times 3$ 2D network, we experimentally recover trajectories of two coupled large spins and demonstrate that the optimization of single-excitation QST yields synchronized precessions among the two mapped spins (Fig. 1e) in the large spin representation. We then adapt our protocol to transfer few particles across a $6 \times 6$ 2D network, a problem for which analytic

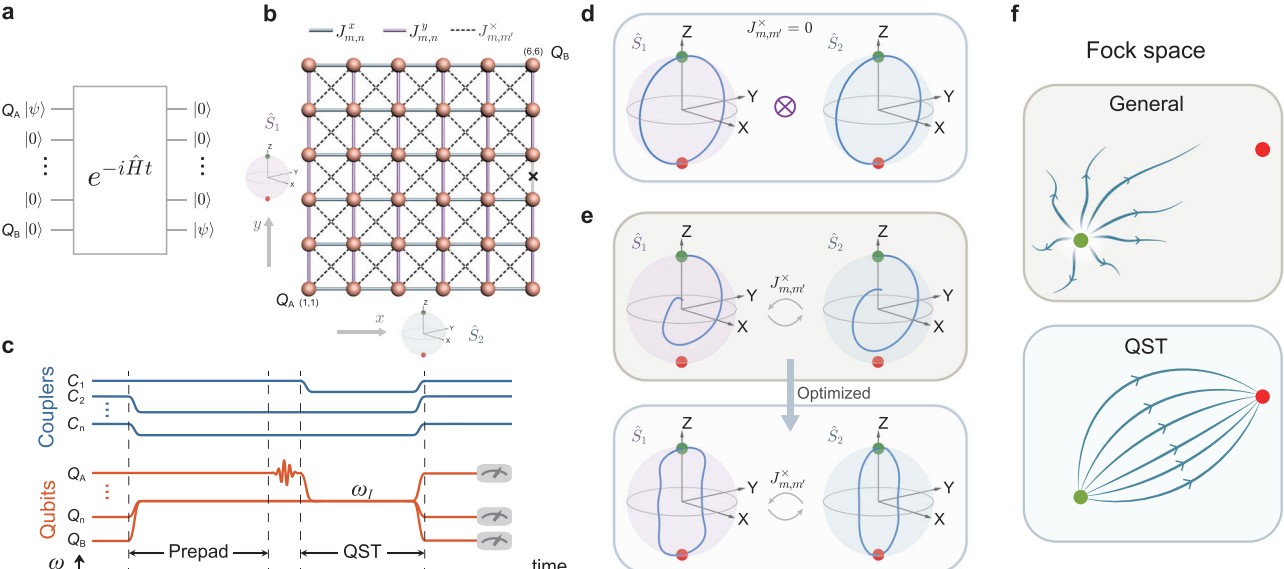

**Fig. 1 | Schematic representation of quantum state transfer. a** Single-excitation QST is achieved by finding a suitable Hamiltonian $\hat{H}$ which transfers initial state $|\psi\rangle$ encoded in qubit $Q_A$ to qubit $Q_B$. Here we assume $\hbar = 1$. **b** Large-spin representation of a QST in a 2D network. Without cross-couplings or defects, a QST from $Q_A$ to its opposite-symmetric qubit $Q_B$ can be regarded as the independent precession of two fictitious spins, each mapping a direction of the qubit network; here $N_1 = N_2 = 6$. NN couplings along the $x(y)$ directions are denoted by $J_{m,n}^x$ ($J_{m,n}^y$), whereas gives the amplitude of the intraplaquette next-nearest neighbor couplings. Gray bond with a cross marker depicts the defect, a malfunctioning coupler in our device. **c** Pulse sequences for realizing single-excitation QST. Square pulses are applied on all other qubits except for $Q_A$ to bring them to the resonant frequency $\omega_I$, and on all couplers non-neighboring to $Q_A$ to engineer them to the desired couplings. To suppress the effects of small pulse distortions caused by step responses, we wait for $2\mu s$ (prepad) before exciting $Q_A$ and bringing it and its neighboring couplers to

target frequencies. After a transfer time $t_{\text{QST}}$, all the qubits, and couplers are tuned to read work points for qubit state measurements. **d** Trajectory $\{\langle \hat{S}_{i,x}(t) \rangle, \langle \hat{S}_{i,y}(t) \rangle, \langle \hat{S}_{i,z}(t) \rangle\}$ in the enlarged Bloch sphere of the two mapped spins, $i = 1, 2$, when the NN couplings are parametrically selected as $J_{n,n+1}^{x \to 1, y \to 2} = J\sqrt{n(6-n)}$, without cross-couplings ($J_{m,m'}^{\times} = 0$) or defect. **e** $J_{m,m'}^{\times} \neq 0$ and defect disturb the perfect precessions, breaking the standard protocol[31], and the desired QST fails. Optimizing couplings $J_{n,n+1}^{1,2}$ compensates for the effects of imperfections, allowing the "wiggled" evolution to achieve QST within desired time scales. **f** Cartoon contrasting the general picture for the evolution in Fock space of an initial state (green dot) under generic or QST-optimized Hamiltonians. General dynamics tend to be ergodic and quickly diffuse the initial information in Fock space, while the QST dynamics manifest nonergodic behavior, re-converging to the final target state (red dot) at later times.

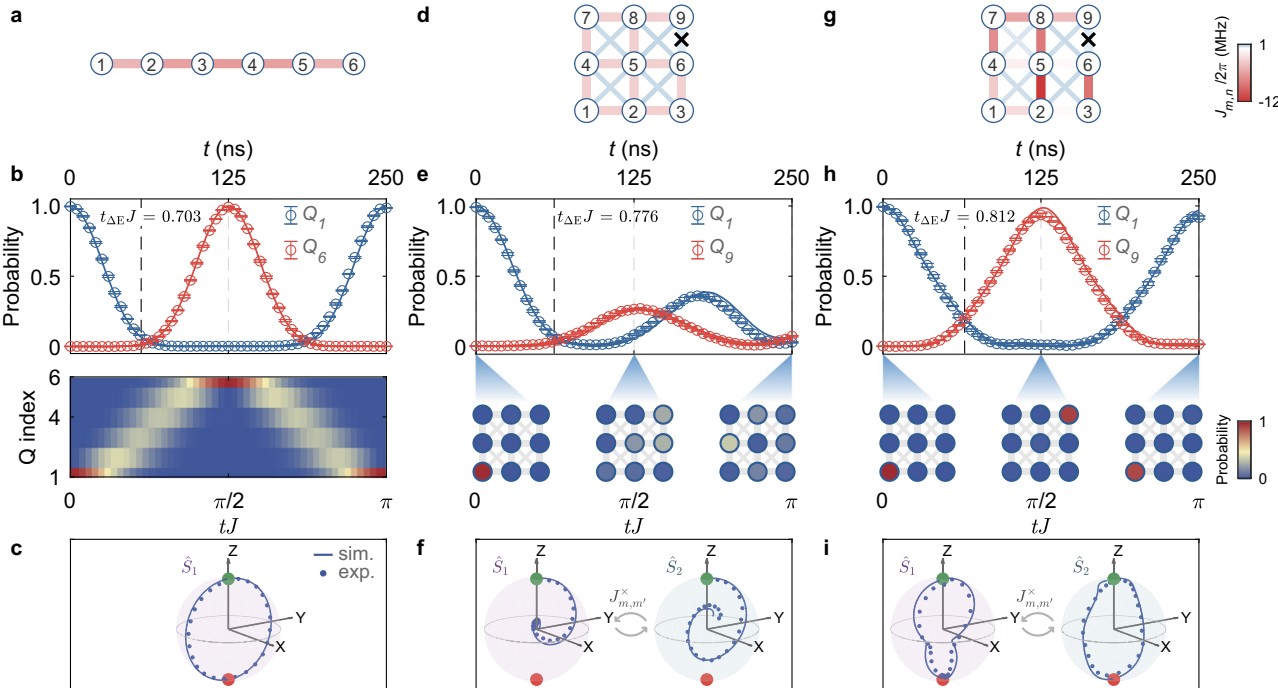

**Fig. 2 | Single-excitation quantum state transfer in 1D (1 × 6) and small 2D (3 × 3) systems. a** Schematic representation of 1D 1 × 6 qubits with the NN couplings parameterized by the standard protocol $J_{n,n+1} = -J\sqrt{n(6-n)}$ with $J/2\pi = 2$ MHz[31]. **b** The corresponding experimental dynamics of $Q_1$ and $Q_6$ excited-state populations (top), whose time-evolution heatmap for all six qubits is shown at the bottom. A nearly perfect transfer fidelity above 0.99 is observed at $t_{QST} \approx 125$ ns. **c** The associated trajectory of $\langle \hat{S}_{1,\alpha} \rangle$ ($\alpha = x, y, z$) in the large-spin representation; markers (line) give the experimental (simulated) results. **d** 3 × 3 qubits with uniform NN couplings, $J_{n,n+1}^{(1,2)}/2\pi \approx -2\sqrt{2}$ MHz[31], except for the defective coupler. **e** The excited population dynamics of qubits $Q_1$ and $Q_9$ (top) and snapshots for all qubits at representative times (bottom). The fidelity is very low (0.27) due to the imperfections. **f** The corresponding spin trajectories, $\langle \hat{S}_{1,\alpha} \rangle$ and $\langle \hat{S}_{2,\alpha} \rangle$, describing the lack of

perfect precession in the presence of cross-coupling terms $J_{m,m'}^{\times}$ and the defective coupler. **g**–**i** show the same for the case in which we optimize the couplings in the 3 × 3 qubit network to achieve a good QST. Despite the wiggled evolution of the spin-trajectories, the synchronized precession is recovered, and the fidelity of the QST is improved to $0.936 \pm 0.012$. The cross marker in the bond connecting qubits $Q_6$ and $Q_9$ denotes the defective coupler with a fixed value of about $2\pi \times 0.3$ MHz. Error bars here come from the standard deviation of five experimental repetitions. $t_{\Delta E}$ is a minimum time, set by "quantum speed limit" arguments, for the generation of a final state localized on a different site, and hence orthogonal to the initial state (see Methods). See Supplementary Fig. 4 for the experimentally measured coupling values in (**a**, **d**, and **g**).

treatments are absent, achieving transfer fidelities of 0.90 for single-excitation, 0.84 for Bell state, and 0.74 for two-excitation, even if cross-couplings and a defective coupler exist. Remarkably, the underlying principle governing a perfect QST of few-particle states is immediately connected with the ergodicity breaking (Fig. 1f) by a near Poisson distribution of the ratio of adjacent gaps. Our findings are far beyond the scope of previous experiments[16,17], not only establishing a practical way to realize few-particle QST in imperfect 2D networks but also revealing the underlying physical understanding of QST from angular momentum theory and quantum ergodicity.

## Results

### Single-excitation transfers

We start by benchmarking the standard one-dimensional protocol of Ref. 31 via employing a single (upper) row of qubits of the current device in Fig. 1b, featuring a 1D chain of $N = 6$ qubits without discernible cross-couplings (Fig. 2a). Figure 2b shows a nearly perfect QST with a fidelity above 0.99 at $t_{QST} \approx 125$ ns ($t_{QST}J \approx \pi/2$) by tuning the qubit couplings $J_{n,n+1} = -J\sqrt{n(6-n)}$ (since $\{J_{n,n+1}\}$ are negative in our experiment, we add a minus sign here to keep the label $J$ as a positive value), with $J/2\pi = 2$ MHz (hereafter we define $J$ as a typical energy scale in our experiments with $J/2\pi = 2$ MHz for 1D and 3 × 3 cases, and 1 MHz for 6 × 6 cases), which maps onto a single large spin precessing around the $x$-axis (Fig. 2c). If a single transfer is desired, the interactions can be switched off at $t_{QST}$ by tuning qubits away from interaction frequency and adjusting NN couplings to the values near zero; otherwise, back-and-forth free propagation of the state occurs

between qubits $Q_1$ and $Q_6$, within time scales (~0.5 μs) such that decoherence ($T_1 \approx 140$ μs, $T_2^{SE} \approx 19$ μs, see Supplementary Note 1) does not substantially affect device performance.

To demonstrate the effectiveness of our procedure, we begin with a case where a defective coupling is deliberately included, and the resulting QST fidelity is low. For that, we explore quantum information transfer in a subset of qubits in Fig. 1b: a 3 × 3 2D network with its lower left corner, $Q_{4,2}$ (see Supplementary Fig. 2), relabeled as $Q_1$ in Fig. 2d, g. It encompasses a device defect – a malfunctioning coupler (the bond with a cross marker in Fig. 2d, g) that constrains one of the qubit couplings to ~$2\pi \times 0.3$ MHz. NN couplings are parametrically calibrated with $J_{n,n+1}^{(1,2)} = -J\sqrt{n(3-n)}$ on other qubit pairs (Fig. 2d). Under these conditions, Figure 2e displays the results of the excited-state population dynamics for $Q_1$ and $Q_9$, quantifying the transfer between a single-qubit excitation initialized at $Q_1$ and aimed to transfer it to the opposite qubit $Q_9$. Unfortunately, the transfer success is largely compromised to a low fidelity of about 0.27 precisely because the existing $J_{m,m'}^{\times}$-couplings and the defect prevent the standard coupling parametrization[31] from achieving perfect QST. In the language of the mapped Hamiltonian, the rotations of the two spins are now correlated, leading to the failure of approaching the pole of the $-z$ direction —see experimental and simulated results in Fig. 2f (see Supplementary Note 5 for details of the measurements of trajectories).

Having shown that parasitic and defective couplings in real devices destroy the transmission of a quantum state with the standard protocol[31], we now tackle these limitations by careful tuning (see Supplementary Note 2) of the coupler-mediated interactions $J_{n,n+1}^{(1,2)}$

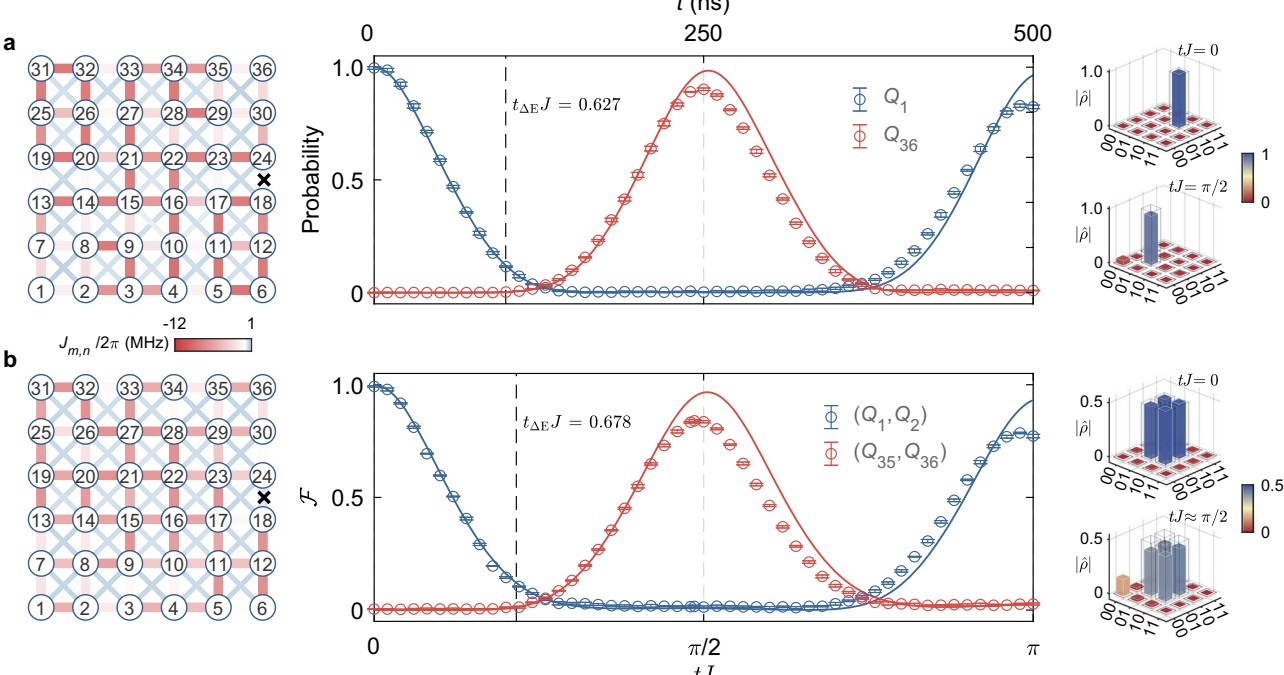

**Fig. 3 | Single-excitation quantum state transfer in a 2D 6 × 6 qubit network with optimized couplings. a** left, shows the measured couplings of the 6 × 6 qubit network; center, the corresponding time evolution of $Q_1$ and $Q_{36}$ excited-state populations, which shows a transfer fidelity of 0.902 ± 0.006 at about 250 ns; right, the quantum state tomography in the subspace of the initial and target qubits, $Q_1$ and $Q_{36}$. **b** Fidelity dynamics for the QST using a Bell state initially encoded in qubits $Q_1$ and $Q_2$; here, the quantum state tomography at $tJ = 0$ is shown in the $(Q_1, Q_2)$ subspace whereas at time $tJ \approx \pi/2$ ($J/2\pi = 1$ MHz) is shown in $(Q_{35}, Q_{36})$ with a fidelity of 0.840 ± 0.006. The fidelity here is a generalization of the probability to the Bell case (see text), where we have two basis states in our initial and final wavefunctions to characterize the QST transfer. Lines (circles) are the numerical (experimental) evolution with the measured couplings. Solid bars (gray frames) represent experimental (ideal) values of density matrix elements. Error bars come from the standard deviation of five experimental repetitions. $t_{\Delta E}$ is the minimum time for a perfect QST set by "quantum speed limit" arguments (see Methods). See Supplementary Figs. 5 and 6 for the specific values of experimentally measured couplings.

(Fig. 2g) according to the couplings optimized with the aforementioned annealing optimization procedure (see Supplementary Note 8). Figure 2h reports these results for the same 3 × 3 network: Regardless of cross-couplings and one fixed coupling, the transfer fidelity is greatly improved, reaching a value of 0.936 ± 0.012. In such a scenario, the trajectories of two coupled large spins $\{\langle \hat{S}_{(1,2),x}(t)\rangle, \langle \hat{S}_{(1,2),y}(t)\rangle, \langle \hat{S}_{(1,2),z}(t)\rangle\}$, are optimized in a way to reach the $-z$ poles synchronously despite their different paths during the dynamics (Fig. 2i). These results pave the way for pursuing QST in much larger quantum circuits, where imperfections are more likely to occur[11,13].

By employing all 36 qubits and utilizing the optimization procedure under those constraints (see Methods), we report in Fig. 3a the transfer of a single excitation across a 6 × 6 qubit network. Here we experimentally achieve a maximum transfer fidelity of 0.902 ± 0.006 (Fig. 3a). Experimentally reconstructed density matrices, labeled by $\hat{\rho}$, of the initial state in $Q_1$ and the resulting final state in $Q_{36}$ after QST are shown in the right panels of Fig. 3a. Under ideal conditions, we can numerically obtain solutions for optimized NN couplings with QST-fidelities above 0.99, even if influenced by cross-couplings and defects (see Supplementary Fig. 18), but experimental imperfections in calibrating couplings and qubit frequencies can impact those results. More prominent, however, are the residual thermal excitations in the qubit network, which mainly cause the observed experimental infidelity. Numerical simulations suggest that 0.5% thermal excitations in each qubit could result in transfer errors of ~3% for 3 × 3 network and ~10% for a 6 × 6 network. This indicates that their suppression constitutes an important route to further improve the future transfer fidelity on a 2D network. For a detailed discussion on the effects of thermal excitations and the noise analysis of couplings and qubit frequencies, see Supplementary Notes 3 and 4, respectively.

Building on these results, our protocol similarly accomplishes the QST of maximally entangled two-qubit states. By preparing a Bell state $|\Psi^-\rangle = (|01\rangle - |10\rangle)/\sqrt{2}$ in qubit pair $(Q_1, Q_2)$, we target the transfer to the opposite-symmetric qubit pair $(Q_{35}, Q_{36})$ in the network (Fig. 3b). For that, the initial state $|\Psi^-\rangle$ is obtained by applying a quantum circuit, which consists of a two-qubit control-Z gate and several single-qubit gates, on $Q_1$ and $Q_2$ (see Supplementary Fig. 14). After a transfer time $tJ \approx \pi/2$, we perform two-qubit quantum state tomography on $Q_{35}$ and $Q_{36}$ to witness the transfer efficiency and find a fidelity $\mathcal{F} = \text{tr}(\hat{\rho}_{\text{exp}}\hat{\rho}_{\text{ideal}}) \approx 0.840 \pm 0.006$ for the experimentally reconstructed density matrix $\hat{\rho}_{\text{exp}}$ (the right panel of Fig. 3b), which demonstrates the effectiveness of our protocol for transferring quantum entanglement. Here, the QST fidelity of the entangled state displays a large sensitivity to noise in both the qubit's frequency and the value of the optimized couplings (see Supplementary Note 4 for an extended discussion), which substantially restricts the transfer success.

**Two-excitation transfers**

The observed relatively large transfer fidelity for states (entangled or not) composed of a single excitation endows the ability to push toward an even more challenging scheme. A standard mapping (see Methods) of the spin operators in Hamiltonian (1) relates the emulated model to one of hardcore bosons, whose cardinality reflects the number of photon excitations and hopping energies $J_{m,n}$. In the case of a 2D lattice, such a model describes a typical quantum chaotic Hamiltonian. Its non-integrability renders quick thermalization and the absence of memory of the initial conditions throughout sufficiently long dynamics[36]. Therefore, the prospects of achieving a successful QST are slim: One expects diffusive behavior in Fock space[37], making it unlikely to find a single state with the majority of the weight in $|\psi(t)\rangle$ at a later time. Yet,

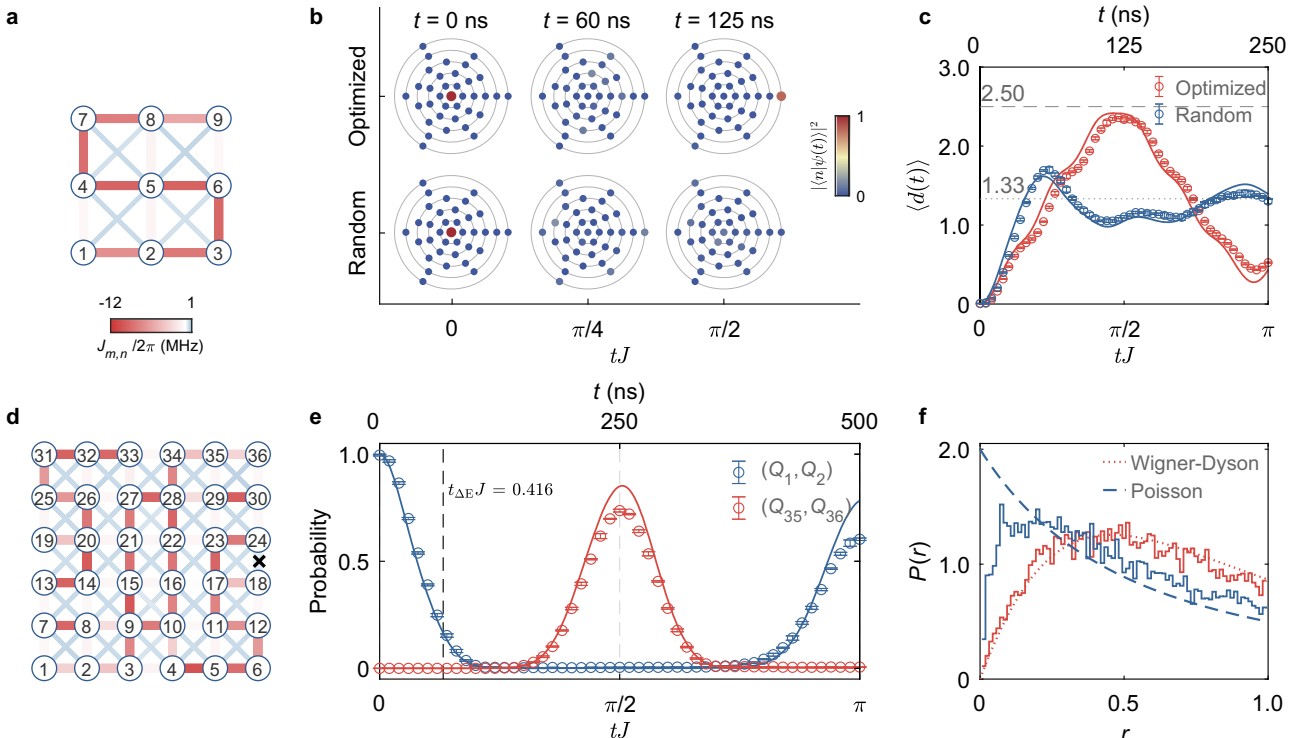

**Fig. 4 | Two-excitation QST in 2D qubit systems with optimized couplings.**
**a** Measured couplings of the 3 × 3 qubit network for the optimized two-excitation QST (see Supplementary Fig. 7 for the specific values). **b** The corresponding experimental time evolution of two-excitation state QST in Fock space [$\mathcal{D}_{\hat{H}} = \binom{9}{2} = 36$], where each marker denotes a Fock state–here, we contrast a solution for QST-optimized couplings from one with randomly chosen $J_{m,n}$ at different representative times. The concentric circles denote the Fock states with the same distance from the initial state. **c** The dynamics of the average distance $\langle d(t) \rangle$ traveled in Fock space for both cases; the dashed (dotted) line gives the maximum (mean) distance. **d** Measured couplings of the 6 × 6 qubit network for the two-

excitation QST [$\mathcal{D}_{\hat{H}} = t\binom{36}{2} = 630$] after optimization (see Supplementary Fig. 8 for the specific values). **e** The $(Q_1, Q_2)$ and $(Q_{35}, Q_{36})$ populations over time using the measured couplings in (**d**), which yield a transfer fidelity of 0.737 ± 0.007 at about 250 ns. Error bars in (**c**) and (**e**) come from the standard deviation of five experimental repetitions. **f** Numerically computed distribution of the ratio of adjacent gaps $P(r)$ in the case of QST-optimized and random couplings. Here, we take an average of an ensemble of $k = 40$ coupling matrices to improve statistics; dashed and dotted lines are surmises for the Wigner–Dyson and Poisson distributions[48], respectively (see Supplementary Note 7). $t_{\Delta E}$ is a minimum time for a perfect QST set by "quantum speed limit" arguments (see Methods).

suppose the number of excitations is small compared to the total system size. In that case, we argue that this weakly chaotic Hamiltonian[38,39] can still be engineered such that a two-excitation QST is efficient (see Supplementary Figs. 20 and 21 for the optimized couplings).

Figure 4a and d show the QST-optimized couplings for two-excitation states in 3 × 3 and 6 × 6 qubit networks–in both cases, the pair $(Q_1, Q_2)$ is initially excited. In the former, we experimentally exemplify in Fig. 4b the propagation of $|\psi(t)\rangle$ in Fock space, contrasting both the optimized and random couplings at different times. Only when the couplings are optimized does one recover a regime where most of the weight collapses on a single Fock state, quantitatively describing the schematic cartoon in Fig. 1f. The states are organized according to a metric defined by the $L_1$-norm of the excitations in the lattice (see Methods) such that the distance $d$ to the initial state $|n=0\rangle$ obeys $d(|0\rangle,|0\rangle)=0$, whereas the distance to the target state, $d(|0\rangle,|n_{\text{target}}\rangle)$, takes the maximum value for the given network size. Thus, an average distance can be dynamically defined as

$$\langle d(t) \rangle = \sum_{n=0}^{\mathcal{D}_{\hat{H}}-1} d(|0\rangle,|n\rangle) |\langle n|\psi(t)\rangle|^2, \qquad (3)$$

measuring the wave-packet's "center of mass" evolution in the Fock space of dimension $\mathcal{D}_{\hat{H}}$. Figure 4c displays its dynamics for the case of the QST-optimized solution of the couplings, where one observes a ballistic (almost) periodic evolution between the initial state $|0\rangle$ and the target state $|n_{\text{target}}\rangle$. Conversely, if one implements random couplings $\{J_{m,n}\}$ between the qubits, a slow evolution towards the

mean distance $\bar{d} = \frac{1}{\mathcal{D}_{\hat{H}}}\sum_{n=0}^{\mathcal{D}_{\hat{H}}-1} d(|0\rangle,|n\rangle)$ ($\bar{d} \approx 1.33$ for 3 × 3 cases) is achieved. In this case, the wave-packet dynamics after an initial transient is close to exhibiting a diffusive behavior (see Supplementary Note 7).

Turning to the large 6 × 6 qubit network, we report in Fig. 4e the population dynamics of the initial and target two-excitation states: Here, a maximum fidelity of about 0.737 ± 0.007 is experimentally observed for the transfer of excitations from $(Q_1, Q_2)$ to $(Q_{35}, Q_{36})$. Notwithstanding the large Hilbert space, $\mathcal{D}_{\hat{H}} = \binom{36}{2} = 630$, one has to deal with in this case to find the optimized couplings that maximize the QST fidelity, the non-integrability of the Hamiltonian naturally bounds performance. Minor deviations on the optimized couplings, which can inherently occur owing to experimental imperfections, significantly impact the transfer fidelity (see Supplementary Note 4).

The approach we have introduced here explicitly uses an annealing Monte Carlo procedure to optimize QST. As such, it is a "black box", providing no clear physical indication of *why* the optimized coupling solutions give a better QST. To acquire that insight, we perform an ergodicity analysis, classifying the eigenspectrum $\{\varepsilon_\alpha\}$ of the corresponding Hamiltonians using the ratio of adjacent gaps $r_\alpha \equiv \min(s_\alpha, s_{\alpha+1}) / \max(s_\alpha, s_{\alpha+1})$[40], where $s_\alpha = \varepsilon_{\alpha+1} - \varepsilon_\alpha$. If the couplings are randomly chosen, Fig. 4f shows that the distribution $P(r)$ typically follows one of the random matrices of the same symmetry class of $\hat{H}$, signifying strong ergodicity in the spectrum where level repulsion takes place [$P(r=0) \to 0$] (see Supplementary Note 7 for an extended discussion). In contrast, if the couplings are optimized, ergodicity is largely absent, and a distribution $P(r)$ close to a Poisson one is obtained

instead. This analysis thus provides a clear understanding of the underlying interpretation of the optimization procedure: the couplings $J_{m,n}$ evolve to partially cure the quantum chaotic nature of the system, allowing higher fidelity QST to take place.

Our Monte Carlo annealing process does not produce a unique solution. Different $J_{m,n}$ can be found which give QST of high fidelity. The ergodicity analysis provides a link between these solutions: They share a distribution of their level spacings, which is roughly Poissonian, with higher fidelities linked to more faithful Poisson statistics.

## Discussion

In this work, we realized efficient QST of few-excitation states in a 2D intermediate-scale quantum processor, even in the presence of unwanted parasitic couplings and inherited defects. The key ingredient relies on manipulating NN inter-qubit couplings, whose values are set by a classical Monte Carlo annealing optimization procedure under the conditions of maximizing the transfer fidelity at a given time $t_{QST}$. Unlike previous single-excitation QST experiments in small-scale 1D chains[16,17], our experiments generalize QST to two dimensions and few-excitation states. More importantly, we reveal the underlying connections between the efficient few-excitation transfer and the breaking of quantum ergodicity.

These demonstrations on an actual physical device underline the significance of our protocol. In practice, unwanted couplings or defects always exist in 1D and 2D qubit arrays. Our protocol not only can be used to realize remote interaction and distribute entanglement across a large solid-state device but also provides a constructive technique for designing quantum channels as building blocks to link two processor nodes[14]. Looking forward, the combination of dual-rail encoding and the emergent mid-circuit measurement technique[41] provide a new avenue to further improve the speed and robustness of QST[42,43].

## Methods

### Monte Carlo annealing process

In our experiments, the intraplaquette cross-couplings $\{J_{m,m'}^{\times}\}$ and the defective coupler cannot be manipulated, while the NN couplings $\{J_{m,n}^{x,y}\}$ for NN qubits $Q_m$ and $Q_n$ are adjustable. The task of accomplishing an efficient QST from the source qubits to the destination qubits relies on an optimization scheme to find appropriate coupling matrices $\{J_{m,n}^{x,y}\}$, in the presence of these constraints. Therefore, we first estimate the cross-coupling values $\{J_{m,m'}^{\times}\}$ and the defect before optimization. Over an extensive experimental calibration process (see Supplementary Note 2), we properly approximate all the next-NN coupling parameters (parasitic couplings) to a value of $J_{m,m'}^{\times}/2\pi = 0.45$ MHz in the optimization. As pointed out in the text, the coupler connecting qubits $Q_{18}$ and $Q_{24}$ is defective, setting the corresponding coupling $J_{18,24}/2\pi$ to a fixed value $+0.3$ MHz. As a result, the optimization process proceeds with these extra constraints.

We employ a Monte Carlo process in this space of parameters with cost function $p(\{J_{m,m+1}^{x,y}, J_{m,m'}^{\times}\}) \propto e^{-\bar{F}/T}$, where the 'temperature' $T$ is varied from $T_{high} \to T_{low}$ and $\bar{F}$ marks the quantity one aims to minimize: the infidelity $\bar{F}(t_{QST}) = 1 - |\langle\psi(t_{QST})|\psi_{target}\rangle|^2$ of the perfect QST at times $t_{QST}$. At each step of the sampling, one obtains such state via unitary time evolution $|\psi(t_{QST})\rangle = e^{-i\hat{H}t_{QST}}|\psi(0)\rangle$, where $\hat{H}$ is constructed with the current couplings parameters $\{J_{m,m+1}^{x,y}, J_{m,m'}^{\times}\}$. Throughout the sampling, we use a combination of local and global parameter changes, combined with $k$ independent realizations of the Monte Carlo process ($k = 40$ for the two-excitation transfer and $k = 5$ for the remaining cases). Furthermore, we also compare different annealing scheduling protocols $f(T)$, where $f(T_{high(low)}) = T_{high(low)}$, and proposed changes in the couplings are dynamically adjusted according to their acceptance ratio.

Among the many choices for inter-qubit coupling matrices that can maximize the fidelity of QST, we focus on the ones that preserve

the network inversion symmetry when dealing with single- or two-excitation transfers. Two reasons stand behind this: (i) such symmetry is also present in the original protocols for perfect QST[31], and (ii) it reduces the space of parameters one needs to probe to find solutions that minimize the infidelity. Under such conditions, we must optimize 30 individual couplings (15 for each direction) on the $6 \times 6$ superconducting quantum circuit instead of 60 couplings if no symmetries were enforced.

Finally, we remark that in a sufficiently large lattice, a possible solution for good QST for multiple excitations is to have disjoint paths along which individual excitations propagate independently. However, we have verified that more complex solutions with equally high fidelity also exist by forcing all the couplings $J_{m,n}$ to be bounded away from the origin. Typically, we have performed the annealing process enforcing that $\{J_{m,n}^{x,y}\}/2\pi \in [J_{min}J_{max}]$ MHz. In general, a larger range yields a higher optimized QST fidelity but becomes more challenging for experimental calibrations. In practice, we set different $J_{min} \in [-12, -6]$ and $J_{max} \in [-0.5, -0.3]$ for different cases (see Supplementary Note 8 for the details of optimized couplings and QST dynamics); ergodicity analysis in Fig. 4f is performed with $\{J_{m,n}^{x,y}\}/2\pi \in [-10, -0.1]$ MHz.

### Cross-couplings

When using the functional form that maximizes QST in a regular lattice, mapping it to a large-spin Hamiltonian, $J_{n,n+1}^{(d)} = J\sqrt{n(N_d - n)}$, possible cross-couplings among the qubits can be incorporated in a similar picture such that $J_{m,m'}^{\times} = J^{\times}\sqrt{m(N_1 - m)}\sqrt{m'(N_2 - m')}$, with $m = 1, ..., N_1 - 1$ and $m' = 1, ..., N_2 - 1$. This is assumed when writing the compact Hamiltonian in Eq. (2). Experimentally, however, active control of these couplings is inaccessible, and as mentioned above, calibration shows that $J_{m,m'}^{\times}$ is approximately constant over the qubit network. As a result, while the final emulated Hamiltonian evades the simple large-spin form in this situation, one can still generically represent and compute the corresponding spin "trajectories," as done in Fig. 1d, even in the presence of non-parameterized cross-couplings.

### Hardcore boson picture

We use the notation of qubit excitations and particles in the text interchangeably. This relies on the standard mapping between hardcore bosons and the spin-1/2 operators: $\hat{a}_m^{\dagger} \leftrightarrow \hat{\sigma}_m^+$ and $\hat{a}_m \leftrightarrow \hat{\sigma}_m^-$[44]. As a result, the Hamiltonian (1) is written as

$$\hat{H} = \sum_{\langle m,n\rangle}^{N} J_{m,n}[\hat{a}_m^{\dagger}\hat{a}_n + \hat{a}_n^{\dagger}\hat{a}_m], \quad (4)$$

where the "couplings" are read as hoppings energies between orbitals $m$ and $n$.

### Distances in Fock space

Given the typical values of the coupling matrix, we define a metric for distances between Fock states inspired by the associated time for a Fock state to be reached. For example, the target state with excitations in the qubit-pair $(Q_{35}, Q_{36})$ in the $6 \times 6$ qubit network should be one of the most distant from the initial state with excitations in $(Q_1, Q_2)$. Using the initial state as a reference, a possible distance is defined as $d(|0\rangle, |n'\rangle) = \frac{1}{4}\sum_{l=1}^{2}(|x'_l - x_0| + |y'_l - y_0| + |x'_l - x_1| + |y'_l - y_1|) - 1/2$, where $(x'_l, y'_l)$ are the Cartesian coordinates of each of the $l$-excitations ($l = 2$) of a generic Fock state $|n'\rangle$. For the initial state $|n = 0\rangle$, one thus have $(x_0, y_0)$ and $(x_1, y_1)$ being the coordinates of its excitations. Hence $d(|0\rangle, |0\rangle) = 0$ whereas $d(|0\rangle, |n_{target}\rangle) = 8.5(2.5)$ for the target state in the $6 \times 6 (3 \times 3)$ network size. The 1/4 prefactor in the definition of $d(|0\rangle, |n'\rangle)$ refers to the average of the four different $L_1$-norm distances to each pair of particles in the two Fock states, $|0\rangle$ and $|n'\rangle$, owing to the particle's indistinguishability.

## Quantum speed limit

The transfer time $t_{QST}$ is bound to obey two constraints: If $t_{QST}$ is large, typical couplings are small in magnitude, but the longer the time, the more drastic the effects of decoherence, which would ultimately inhibit an efficient quasi-adiabatic QST. On the other hand, a fast QST is bounded by fundamental limits of the evolution of *any* quantum mechanical system. Dubbed quantum speed limits, they control the minimal time scale necessary for a time-evolving wave function to become fully distinguishable (i.e., orthogonal) from the initial state. Such a perfect orthogonalization process precisely describes a flawless QST. Known bounds[45,46] limit the minimal orthogonalization time based on either the mean energy $E = \langle \hat{H} \rangle$ or the energy uncertainty $\Delta E = \sqrt{\langle \hat{H}^2 \rangle - \langle \hat{H} \rangle^2}$ of the system

$$t_{\Delta E} = \frac{\pi \hbar}{2 \Delta E}, \quad t_E = \frac{\pi \hbar}{2(E - E_g)}, \quad (5)$$

with $E_g$ the ground-state energy of the Hamiltonian $\hat{H}$ that governs the unitary dynamics. As a result, $t_{QST} \geq \max\{t_{\Delta E}, t_E\}$. In Figs. 2, 3, and 4, we include the minimal orthogonalization time given by the quantum speed limit for each of the Hamiltonians that describes the corresponding evolution (see Supplementary Note 9 for an extended analysis). In all cases we investigate, the energy uncertainty limit bounds the QST, i.e., $t_{QST} \geq t_{\Delta E}$.

## Data availability

The data generated in this study have been deposited in the Zenodo database under accession code https://doi.org/10.5281/zenodo.11090630[47].

## Code availability

The codes used for Monte Carlo optimization in this study are available in the Code Ocean capsule at https://codeocean.com/capsule/8181086/tree/v2.

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

## Acknowledgements

We thank Siwei Tan and Liangtian Zhao for their technical support. RM acknowledges insightful discussions with David Weiss, Marcos Rigol, and Pavan Hosur. The device was fabricated at the Micro-Nano Fabrication Center of Zhejiang University. This research was supported by the National Natural Science Foundation of China (Grant Nos. 92065204, NSAF-U2230402, 12222401, 11974039, U20A2076, 12274368), the Zhejiang Province Key Research and Development Program (Grant No. 2020C01019). RTS was supported by the grant DE-SC0014671 funded by the U.S. Department of Energy, Office of Science. QG was also supported by the Zhejiang Provincial Natural Science Foundation of China (Grant Nos. LR24A040002, LQ23A040006) and the Zhejiang Pioneer (Jianbing) Project (Grant No. 2023C01036).

## Author contributions

R.M. and R.T.S proposed the idea; A.Y., J.P., Z.Z., and R.M. performed the numerical simulations; L.X. and Z.Z. conducted the experiment under the supervision of Q.G. and H.W.; J.C and H.L. designed and fabricated the device under the supervision of H.W.; R.M., Q.G, and R.T.S. co-wrote the manuscript; H.W., Q.G., C.S., Z.W., L.X., J.C., Z.S., Z.B., Z.Z., X.Z., F.J., K.W., S.X., and Y.Z. contributed to the experimental setup.

## Competing interests

The authors declare no competing interests.
