## [Peer Review File · Nature Communications]

Enhanced quantum state transfer by circumventing quantum chaotic behaviorREVIEWER COMMENTS

Reviewer #1 (Remarks to the Author):

The manuscript “Enhanced quantum state transfer: Circumventing quantum chaotic behavior” by L. Xiang and co-authors is a nice mixture of experimental, theoretical and numerical results on quantum state transfer in a mid-size superconducting quantum processor up to 36 qubits.

The main technological enabler on the experimental side is the fact that the coupling/hopping strength between the qubits is tunable (which as such is not a new achievement). The theoretical primer idea originating in Refs. 37 & 39 is that by organizing the qubit-qubit coupling strengths proportional to the geometric mean distance from the initial n and final locations $(N-n)$, $\sqrt{n(N-n)}$, one achieves perfect state transfer in an otherwise ideal situation.

To go beyond the known results and to cope with imperfections such as malfunctioning couplers (denoted as a defect in the manuscript) and additional non-tunable cross-couplings, the authors have numerically simulated the quantum state transfer and searched for optimal qubit-qubit coupling strengths with well-matching experimental results. One of the highlights of the manuscript is that the authors have neatly connected the optimal quantum state transfer in the complex non-ideal 2D qubit grid with the theory of non-ergodic vs. ergodic Hamiltonians via quantum many-body energy level statistics. The main intuition they offer is that it seems that for optimal quantum state transfer one needs to have non-ergodicity to avoid diffusive behavior in Hilbert space.

The manuscript is a pleasure to read and it is well understandable even on the first read. It has a coherent and logical structure with clear writing (see below some suggestions for improvements).

The results have a good potential to have considerable impact in better design of solid-state quantum processors, improved state transfer protocols and in enhancing the connection between quantum many-body physics and applied quantum information topics. However,

we have some criticisms and comments (see below) regarding, among other things, connection of this work to earlier studies of quantum information propagation in superconducting quantum processors and other demonstrations of long-range quantum operations. Thus, we feel sorry these concerns preclude us immediately recommending publication in Nature Communications.

Major comments:

- The manuscript doesn't reference enough other recent studies on quantum information propagation in small-to-mid-size quantum processors. For example, A. Karamlou et al., *npj Quantum Inf.* 8, 1 (2022); J. Braumüller et al., *Nat. Phys.* 18, 172 (2022). Could you please extend the discussion in the introduction on these parts?

-In A. Karamlou et al., *npj Quantum Inf.* 8, 1 (2022), they demonstrate transport of an excitation over a distance of 6 sites with a fidelity that looks almost perfect (see Fig 1 d there). This is achieved with constant coupling strengths J between the qubits. Can you comment how this simple uniform coupling scheme relates to your coupling scheme based on the geometric mean distance scaling or the numerically optimized scheme? Can one achieve perfect (or almost perfect) state transfer this way too? Based on Ref [37] it seems that a constant coupling scheme is insufficient when the length is more than 4 sites. Can you give some intuition behind this?

-The manuscript doesn't provide a comparison of the presented method to the alternative SWAP gate method for quantum state transfer. Which of the methods is faster (taking into account that there are some practical limits for the achievable coupling strengths between qubits) or more accurate or more scalable? Furthermore, just out of pure curiosity, the recent arXiv manuscript by E. Bäumer et al., arXiv:2308.13065, compares the SWAP method and a quantum teleportation based dynamic method involving mid-circuit measurements and feed-forward operations for teleporting two-qubit gates [whose less difficult version is state teleportation]. Can you comment how your method compares to the presented teleportation method?

- How does the numerical optimization method based on Monte Carlo annealing process

scale as a function of the qubit grid?

- Context of the results: Some more context and relevance for the results would be needed in relation to others and the same platform. For example, you cite the reference [7] that uses superconducting quantum circuits featuring four qubits with similar purposes. It would be important to mention more clearly what is new in your research, is it 2D, cross-coupling or the defect?

- Role of the inversion symmetry: You say that you use inversion symmetry as a restriction to the optimization algorithm following the idea of previous work performing perfect QST in 1D. But later (e.g. in Supplementary section 8), the inversion symmetry is not utilized. Could you please elaborate on this a bit more deeply? What is its relevance or is it just a way of reducing the space of the parameters? Does inversion symmetry play a central role here? I think it could be interesting to explain this better, i.e. the relationship between inversion symmetry, spatial dimensions, number of excitations, and the presence/absence of defects. Furthermore, what is it known analytically?

- Rewriting of Discussion: In the two paragraphs, the first one is lengthy and delves into the theoretical speed limit of the system. As there is no explicit earlier mention throughout the main text of the speed limits, maybe including these details in the Method Section/or and providing some comments in the Result section would work better? Furthermore, the Discussion should in general include some more comments about the results, their relation with previous research, and possible future impact.

Minor comments:

Introduction:

- Is “platform-agnostic” of standard use in the quantum technology community? Maybe using platform-independent could be clearer for general readers.

- Please try to keep the notation for the sites coherent: Eq. (1) starts with “ij” and then you use “n”. For example, instead of $J_{\{i,j\}}$ and $J_{\{n,n+1\}}$, you could simply use $J_{\{i,j\}}$ and $J_{\{i,i+1\}}$ and additionally denote that $J_{\{i,i+1\}}$ refers now to 1D chain.

- The introduction section is quite long, almost 3 pages, covering both general introduction and the theoretical background. It might be useful to restructure it for better readability. Furthermore, it might be useful to have a more concrete text part explaining the structure of the paper. It is now explained quite briefly in the last part of the introduction on page 3.
- The sentence “establishing a unified and fundamental understanding of perfect QST from angular momentum theory and quantum ergodicity” could be more direct and explicit about the main findings of the manuscript, or you could include a deeper elaboration of it in the discussion section.

Results:

- What does the sentence “the remaining qubits can be switched off at t_{QST} ” mean exactly? How is it implemented?
- You could add to the sentence “within time scales such that decoherence does not substantially affect device performance” some order of magnitude. Although this information is in the Supplementary material, it is a bit hidden.
- It is not clear what experimental parameters you can manipulate and to what extent. If I understand correctly, you measure the cross-coupling values $J_{\{m,m'\}}^x$, which cannot be manipulated. Subsequently, you introduce this value into the optimization algorithm to calculate the NN couplings $J_{\{i,j\}}$, which are adjustable. Is my understanding correct? You could explain this in the Method Section more clearly.
- The references to the Supplementary Sections could include the exact supplementary figures you are referring to.
- I do not understand the necessity of introducing the new distance metric for two excitations instead of using Fidelity, as you used in the previous cases. Could you please provide a more detailed explanation? By the way, is this equal or proportional to the so-called Manhattan distance?

- What does the term 'pattern' mean in the sentence "there is no discernible pattern in the optimized J_{ij} , unlike the case of the 1D protocol."? Is it not related to mirror (inversion) symmetry? For instance, the circuit depicted in Figure S19 exhibits a certain pattern in this sense. Could you clarify what you mean exactly here?

Generally:

-The manuscript uses a notation where operators have hats, then to be consistent, also the density matrix should have a hat in Figures and in the text.

Reviewer #2 (Remarks to the Author):

The manuscript seems of high quality, well motivated and written, easily accessible, good plots. Generally, the manuscript thus seem suitable for NAT. Comm.

The results are also of high technical quality: Large number of participating qubits (36), high fidelities of operations, careful optimization of pulse parameters.

Two points for potential improvements:

1) The concepts and results described in the main text are disconnected from the actual implementation. To improve on this I would move plots such as S11 from the supp mat to the main text

2) It is good practice to provide information about the quantum device. Very little is said about it. This is important to relate the infidelities to properties of the quantum hardware.

Reviewer #3 (Remarks to the Author):

The authors previously demonstrated the quantum simulation on the 1D superconducting qubit system [36] based on the theoretical proposal for quantum state transfer on a 1D XY-spin chain [4]. In this paper, they demonstrated the similar experiment on the 2D superconducting qubit system based on the theoretical extension into the multi-

dimensional system [40]. The extension from 1D to 2D has implications beyond mere complexity: while 1D systems are integrable, 2D systems with multiple excitations are known to cause chaos, and the required accuracy of the experiments is expected to be high. The authors argue that it is difficult to apply ideal state transfer as in real quantum processors due to the imperfections of the experimental system, and they circumvent this problem by optimizing the coupling strength between qubits using Monte Carlo annealing.

Their system is a square lattice of 6×6 data qubit systems mediated by coupler qubits. Both the data and the coupler qubits are frequency-tunable, effectively allowing the frequency relationship and the coupling strength between the data qubits to be freely manipulated. The lifetime, coherence, readout, and two-qubit gate fidelity in specific qubit pairs for bell-state preparation are described in the appendix 1. In the demonstration, they first prepared the several initial states at the one end of the system, such as single-excitation state, bell state, and two-excitation states. They next transfer these states to the other end of the system within 500 ns. They finally evaluate the state transfer fidelity as 0.90, 0.84, and 0.74, respectively, with the quantum state tomography. In the appendix 3 and 4, they analyze the error source of the state transfer in their experiments. They declare two factors as the possible main error sources: residual thermal excitation after initialization of the qubits (results to $\sim 10\%$ error) and estimation errors in the system parameters (results to tens of % error).

In the discussion, they numerically analyze how the optimized Hamiltonian differs from the Hamiltonian with the random coupling strength. They show the while the Hamiltonian with the random coupling strength show the ergodic feature, the optimized Hamiltonian show the nonergodic feature through their statics of the level spacing distribution. They estimate that their optimization of the coupling strength suppresses chaos and thus increases the state transfer fidelity.

This study has two aspects: quantum simulation of multidimensional XY-spin system and model-based quantum optimal control on the large-scale superconducting qubit system. The proposed method of optimizing state transfer using Monte Carlo annealing is scientifically interesting, but as described in the major revisions below, we are not confident

of its practicality. On the other hand, their demonstration shows that they comprehensively understand and control the details of the system, including its tunability and incompleteness. Therefore, we support the acceptance of this manuscript if they address the major revisions listed below.

Major revisions:

1. In the Monte Carlo annealing, they calculated the state transfer infidelity as a cost function by simulating the time evolution of a large-scale quantum system on a classical computer. It clearly does not scale. Is there any prospect to make it scalable?
2. Is it possible to visualize the final state of all qubits in the system, not just the initial and final qubits of Fig3 and 4? This information may be helpful to the reader for considering state transfer error sources.
3. If the state transfer time is extrapolated beyond π , do you see repeated state transfers between the start and end points?
4. While the proposed method allows state transfer via multiple paths, it does not necessarily result in the fastest state transfer as it does not necessarily employ the maximum coupling achievable in the system. It is useful to roughly compare the state transfer speed with the case of sequential SWAP (or iSWAP) gates.
5. The author considers only diagonal qubits as NNN coupling, but is there an estimate of the strength of NNN coupling between two adjacent qubits like Q1-Q3?
6. Have you measured and corrected the flux crosstalk in the system? If crosstalk remains, it may be a source of estimation errors in the qubit frequency and the coupling strength between qubits. Also, please describe how much estimation error the authors expect for the system parameters in appendix 4.

Minor revisions:

1. There is a previous study [A] regarding quantum walks on two-dimensional superconducting qubit system. It would be good to be cited in the main text.
2. Please add state transfer execution time to Figure 2.
3. Please refer the single qubit gate fidelity of the system in the appendix 1.

[A] M. Gong, et al., Science 372, 948-952 (2021).

Point-by-Point Response

Response to Reviewer # 1

Comment 1.0: The manuscript “Enhanced quantum state transfer: Circumventing quantum chaotic behavior” by L. Xiang and co-authors is a nice mixture of experimental, theoretical and numerical results on quantum state transfer in a mid-size superconducting quantum processor up to 36 qubits.

The main technological enabler on the experimental side is the fact that the coupling/hopping strength between the qubits is tunable (which as such is not a new achievement). The theoretical primer idea originating in Refs. 37 & 39 is that by organizing the qubit-qubit coupling strengths proportional to the geometric mean distance from the initial n and final locations ($N-n$, $\sqrt{n(N-n)}$), one achieves perfect state transfer in an otherwise ideal situation.

To go beyond the known results and to cope with imperfections such as malfunctioning couplers (denoted as a defect in the manuscript) and additional non-tunable cross-couplings, the authors have numerically simulated the quantum state transfer and searched for optimal qubit-qubit coupling strengths with well-matching experimental results. One of the highlights of the manuscript is that the authors have neatly connected the optimal quantum state transfer in the complex non-ideal 2D qubit grid with the theory of non-ergodic vs. ergodic Hamiltonians via quantum many-body energy level statistics. The main intuition they offer is that it seems that for optimal quantum state transfer one needs to have non-ergodicity to avoid diffusive behavior in Hilbert space.

The manuscript is a pleasure to read and it is well understandable even on the first read. It has a coherent and logical structure with clear writing (see below some suggestions for improvements).

The results have a good potential to have considerable impact in better design of solid-state quantum processors, improved state transfer protocols and in enhancing the connection between quantum many-body physics and applied quantum information topics. However, we have some criticisms and comments (see below) regarding, among other things, connection of this work to earlier studies of quantum information propagation in superconducting quantum processors and other demonstrations of long-range quantum operations. Thus, we feel sorry these concerns preclude us immediately recommending publication in Nature Communications.

Response 1.0: We thank the Reviewer for highlighting that our manuscript is “a nice mixture of experimental, theoretical and numerical results” and has “a good potential to have considerable impact in better design of solid-state quantum processors, improved state transfer protocols and in enhancing the connection between quantum many-body physics and applied quantum information topics”. We also appreciate their constructive comments for further improving our manuscript. We believe the improved version, addressing the Reviewer’s comments, is suitable for publication in Nature Communications.

Major comments:

Comment 1.1: The manuscript doesn’t reference enough other recent studies on quantum information propagation in small-to-mid-size quantum processors. For example, A. Karamlou et al., *npj Quantum Inf.* 8, 35 (2022); J. Braumüller et al., *Nat. Phys.* 18, 172 (2022). Could you please extend the discussion in the introduction on these parts?

Response 1.1: Good suggestion, and we thank the Reviewer for reminding us of these studies. These two references, *npj Quantum Inf.* 8, 35 (2022) and *Nat. Phys.* 18, 172 (2022), are now cited in the introduction. We also acknowledged their contribution to exploring quantum transport in two-dimensional networks.

Comment 1.2: In A. Karamlou et al., *npj Quantum Inf.* 8, 1 (2022), they demonstrate transport of an excitation over a distance of 6 sites with a fidelity that looks almost perfect (see Fig 1 d there). This is achieved with

constant coupling strengths J between the qubits. Can you comment how this simple uniform coupling scheme relates to your coupling scheme based on the geometric mean distance scaling or the numerically optimized scheme? Can one achieve perfect (or almost perfect) state transfer this way too? Based on Ref [37] it seems that a constant coupling scheme is insufficient when the length is more than 4 sites. Can you give some intuition behind this?

Figure 1: Quantum state transfer in a 7-site chain of constant couplings. **a**, Schematic of a 7-site 1D chain. The neighboring qubits are coupled by a constant coupling J . **b**, Heatmap of numerically simulated population (P_1) dynamics for a single excitation initialized at Q_1 , which reproduces the Fig. 1d in *npj Quantum Inf.* 8, 35 (2022). **c**, Population dynamics of Q_1 and Q_7 ; same data as in **b** but selecting the qubits at the ends of the chain. The maximum state transfer fidelity from Q_1 to Q_7 within 25 hopping times is ~ 0.882 . **d**, Ordered eigenenergies of the system in single-excitation Hilbert subspace. **e**, Energy level spacings $\{E_{n+1} - E_n\}$. The level spacings are not uniform and thus lead to an insufficient QST.

Response 1.2: In *npj Quantum Inf.* 8, 35 (2022), the authors neither achieve nor claim a near-perfect QST with a constant coupling; data visualized with 2D heatmaps can often be misleading. As shown in Fig. 1 above, we numerically reproduce their results in the same case, a 1D chain with 7 sites (Fig. 1a). Although it looks as if a near-perfect quantum state transfer (QST) occurs in the 2D heatmap (Fig. 1b), the true transfer fidelity is only about 0.882 at short-time scales (Fig. 1c).

As pointed out in *Phys. Rev. Lett.* 92, 187902 (2004), one can not realize a perfect QST in a homogeneously coupled chain with $N \geq 4$ sites. The intuition behind this can be understood from the perspective of the eigenenergies (E_n). Oscillations of quantum dynamics are governed by energy level spacings, $\{E_{n+1} - E_n\}$, of the Hamiltonian \hat{H} . A perfect QST from a site A ($|\psi_A\rangle = \sum_n A_n |E_n\rangle$) to a site B ($|\psi_B\rangle = \sum_n B_n |E_n\rangle$) means a set of eigenstates exactly oscillate to another set of eigenstates *simultaneously*. In other words, the energy level spacings must have strict periodicity. For a spin chain with symmetric couplings ($|J_n|^2 = |J_{N-n}|^2$), the necessary

and sufficient conditions for a perfect QST is

$$E_n - E_{n-1} = (2m_n + 1)\pi/t_{\text{QST}}, \quad (1)$$

where $\{E_n\}$ is an ordered set of eigenenergies with $E_n < E_{n+1}$, t_{QST} is the transfer time, and m_n is an arbitrary integer [see equation (4) in *Phys. Rev. A* 84, 022311 (2011)]. For a constant coupling scheme, the eigenenergy $E_n = -2J \cos(\frac{n\pi}{N+1})$. For $N = 3$, the eigenenergies are $\{-\sqrt{2}J, 0, \sqrt{2}J\}$ with a uniform spacing $\sqrt{2}J$. However, for $N \geq 4$, the eigenenergies are not equal spacing anymore and fail to realize a perfect QST. Figures 1d and e display eigenenergies and level spacings for the 7-site uniform chain in Fig. 1a, where equation (1) is not satisfied thus there is no perfect QST. In contrast, Christandl's scheme, $J_{n,n+1} = J\sqrt{n(N-n)}$, always yields an uniform level spacing of $2J$.

Lastly, it's important to emphasize that equidistance of the energy levels is a sufficient but not necessary condition for *efficient* QST. The numerical solutions we obtain after the optimization (which do not have equal energy gaps) show an example of that.

Comment 1.3: The manuscript doesn't provide a comparison of the presented method to the alternative SWAP gate method for quantum state transfer. Which of the methods is faster (taking into account that there are some practical limits for the achievable coupling strengths between qubits) or more accurate or more scalable? Furthermore, just out of pure curiosity, the recent arXiv manuscript by E. Bäumer et al., arXiv:2308.13065, compares the SWAP method and a quantum teleportation based dynamic method involving mid-circuit measurements and feed-forward operations for teleporting two-qubit gates [whose less difficult version is state teleportation]. Can you comment how your method compares to the presented teleportation method?

Figure 2: Quantum circuits for transferring single excitation in a 1×6 chain using different types of two-qubit gates. **a**, Five consecutive SWAP gates are applied for transferring an excitation from Q_0 to Q_5 . **b-d**, By compiling the SWAP gate into two-qubit gates (e.g., CZ, iSWAP, ECR) and single-qubit gates, we can obtain the quantum circuits in **b**, **c**, and **d**.

Response 1.3: The Reviewer raised a good question. To answer this question in the settings of real devices, we perform numerical simulations for the QST realized with digital quantum gates. In practice, iSWAP-like [*Nature* 574, 505 (2019), Google], controlled-Z [CZ, *Nature* 614, 676 (2023), Google] and echoed cross-resonance [ECR, *Nature* 618, 500 (2023), IBM] gates have been demonstrated with high fidelity on multi-qubit superconducting processors. Therefore, we use parameters in these papers, including gate errors, gate length and coherence times, to perform numerical simulations. The results for different cases are summarized in Ta-

Source	Pauli errors	T_1 T_2	Gate length	Fidelity of 1×6 single excitation / t_{QST}	Fidelity of 6×6 single excitation / t_{QST}	Fidelity of 6×6 Bell state / t_{QST}	Fidelity of 6×6 two excitation / t_{QST}
This paper	-	142 us 19 us	Not applied	0.992 125 ns	0.902 250 ns	0.840 250 ns	0.737 250 ns
Google Nature 574, 505 (2019)	SQ: 0.16e-2 iSWAP: 0.62e-2	16 us 30 us	SQ: 25 ns iSWAP: 12 ns	0.978 60 ns	0.961 120 ns	0.955 108 ns	0.929 108 ns
Google Nature 614, 676 (2023)	SQ: 0.1e-2 CZ: 0.6e-2	20 us 30 us	SQ: 34 ns CZ: 25 ns	0.953 590 ns	0.909 1180 ns	0.897 1062 ns	0.838 1062 ns
IBM (kyiv) Nature 618, 500 (2023)	SQ: 6.75e-4 ECR: 1.15e-2	293 us 157 us	SQ: 50 ns ECR: 612 ns	0.941 6620 ns	0.889 13240 ns	0.848 11916 ns	0.808 11916 ns

Table 1: Numerical results of QST with digital quantum gates. We use IBM Qiskit to simulate the noisy circuits for QST and calculate the transfer fidelity. Examples of quantum circuits for transferring single-excitation in a 1×6 chain are shown in Fig. 2. The parameters to build the error models, including qubit coherence times T_1 , T_2 , the mean Pauli errors of single-qubit gates (SQ) and two-qubit gates (iSWAP, ECR and CZ) are obtained from the previous literature.

ble 1, and the quantum circuits for single-excitation QST in a 1×6 chain are plotted in Fig. 2. Two pieces of information can be obtained from the simulation results.

(1) Fidelity. For the QST in the 1D chain, whose Hamiltonian is integrable, our paper has the best fidelity among all the cases in Table 1, which means a simple scale-up scheme by repeating the 1×6 structure to realize high-fidelity QST in larger system sizes. However, for the imperfect 2D network with cross-couplings and defects, the comparison depends on the gate type and gate fidelity. The transfer with Google’s iSWAP gate in *Nature 574, 505 (2019)* yields the best fidelity because it has both high gate fidelity and short circuit depth. In contrast, the case with IBM’s ECR gate in *Nature 618, 500 (2023)* only apparently outperforms our scheme for two-excitation QST, where the system is non-integrable.

(2) QST time. The transfer time using digital gates largely depends on gate type and gate length. Google’s iSWAP gate in *Nature 574, 505 (2019)* has the shortest transfer time among all the cases, since it has the shortest gate length and could realize a swap operation using only one two-qubit gate. On the contrary, other cases (CZ and ECR) are far slower than our scheme. If considering a more fair comparison, where the maximum coupling of the system is J_{max} , single-excitation QST in an N -qubit 1D chain with consecutive NN swap operations needs a transfer time $t = \frac{\pi(N-1)}{2J_{\text{max}}}$. For the pre-engineered couplings, $J_{n,n+1} = J\sqrt{n(N-n)}$, J_{max} appears at the middle of the chain, $J_{\text{max}} = JN/2$ (assume N is an even number). Thus, the QST time is $t = \frac{\pi N}{4J_{\text{max}}}$, about 2 times faster than the swap method.

As for the quantum teleportation with mid-circuit measurements and feed-forward operations, the main advantage is that it only requires a constant depth of quantum gates to spread quantum information. We use the dynamic circuits in *arXiv:2403.18768* to implement QST (see Fig. 2 for the dynamic circuits for transferring single-excitation in a 1×6 chain) and the numerical results are summarized in Table. 2. Due to the longer readout time and larger readout errors (1 order of magnitude larger than unitary gate operation) for current superconducting processors, the dynamic circuit method still doesn’t show advantages in transferring quantum states in most of cases.

Comment 1.4: How does the numerical optimization method based on Monte Carlo annealing process scale as a function of the qubit grid?

Response 1.4: That is a very good suggestion that prompted us to include an extra Section in the Supple-

Figure 3: Dynamic circuits for single-excitation QST in a chain of 6 qubits. a, Quantum circuit for transferring single excitation from Q_0 to Q_5 , using IBM’s ECR gates as the two-qubit entanglement gate. **b,** Same circuit as **a**, but using CZ gates as the primitive two-qubit entanglement gates provided by Google’s Sycamore processor.

Source	Total length of readout and feedforward	Readout error	Idle error during readout and reset	Fidelity of 1×6 single excitation / t_{QST}	Fidelity of 6×6 single excitation / t_{QST}	Fidelity of 6×6 Bell state / t_{QST}	Fidelity of 6×6 two excitation / t_{QST}
This paper	-	-	-	0.992 125 ns	0.902 250 ns	0.840 250 ns	0.737 250 ns
IBM (dynamic circuit) Nature 618, 500 (2023) & arXiv:2308.13065	2234 ns	1.53e-2	0.9e-2	0.921 3858 ns	0.875 3858 ns	0.640 3858 ns	0.755 3858 ns
Google (dynamic circuit) Nature 614, 676 (2023)	660 ns	2.0e-2	2.5e-2	0.914 903 ns	0.861 903 ns	0.617 903 ns	0.726 903 ns

Table 2: Numerical results of QST with dynamic circuits. Numerical simulations are performed using IBM Qiskit. Examples of dynamic circuits for transferring single-excitation in the 1×6 chain are shown in Fig. 3. For IBM’s processor, the time for qubit readout, feedforward control, and reset operation is estimated as 3.65 times the length of a two-qubit gate [see *arXiv:2308.13065*]. For Google’s processor, the time for feedforward operation is currently unavailable. Therefore, we only count for the total time of readout length (500 ns) and reset operation (160 ns), summing up to 660 ns. Other parameters for gate errors, T_1 , T_2 , and gate length are the same as Table 1.

mentary Materials [Section 10]. We will summarize it here for completeness. Numerous possible protocols exist to achieve optimization under the constraints of minimization of a cost function. Our original idea was that instead of focusing on elaborate approaches, we tackle the easiest possible method, a simple Monte Carlo annealing process in the space of the coupling parameters, as originally described in the Methods section. The main bottleneck in this idea is to efficiently compute the infidelity $\tilde{F}(t_{\text{QST}}) = 1 - |\langle \psi_{\text{target}} | \psi(t_{\text{QST}}) \rangle|^2 = 1 - |\langle \psi_{\text{target}} | e^{-i\hat{H}'t_{\text{QST}}} | \psi(0) \rangle|^2$ for a given Hamiltonian \hat{H}' that emerges in the sampling and necessary to decide whether to accept or reject it in the Monte Carlo process. To accomplish that in a direct approach, one performs the eigendecomposition of the unitary evolution operator, $e^{-i\hat{H}'t_{\text{QST}}} = \hat{U} e^{-i\hat{D}'t_{\text{QST}}} \hat{U}^\dagger$, an operation whose computational complexity scales as $\mathcal{O}(N^3)$ [$N = \binom{N_{\text{qubits}}}{N_{\text{exc}}}$]. While this sets a fundamental restriction in the optimization (another factor proportional to N_{qubits} emerges because of the sampling process), one should recall that the sparseness of the Hamiltonian can be leveraged to make it a straightforward operation. In particular for obtaining $|\psi(t_{\text{QST}}) = e^{-i\hat{H}'t_{\text{QST}}} |\psi(0)\rangle$, one does not need to compute the (usually dense) matrix $e^{-i\hat{H}'t_{\text{QST}}}$, but rather to compute the action of this matrix on a vector. There are efficient methods to accomplish this task, such as the Krylov method – see <https://slepc.upv.es/documentation/reports/str7.pdf> for details and Chapter 7 in <https://slepc.upv.es/documentation/slepc.pdf> for details of the specific procedure we employ. Utilizing this, we show in the revised Supplementary Materials that high-fidelity quantum state transfer can also be obtained in the case of 32×32 qubit networks (i.e., 1024 qubits) for single-excitations or 8×8 cases with two-excitations (here with $N = 2016$). All of that with a computational cost that is comparable with the direct eigendecomposition approach in much smaller system sizes.

Apart from the technical aspects regarding numerical efficiency for large cases, the fact that we can easily obtain solutions with this simple optimization algorithm for substantially large system sizes points to the conclusion that effects of ‘barren plateaus’ are not yet substantial for this specific problem, thereby indicating that quantum state transfer of few-particles is an optimization-efficient process.

Comment 1.5: Context of the results: Some more context and relevance for the results would be needed in relation to others and the same platform. For example, you cite the reference [7] that uses superconducting quantum circuits featuring four qubits with similar purposes. It would be important to mention more clearly what is new in your research, is it 2D, cross-coupling or the defect?

Response 1.5: Thanks for the constructive suggestion. *Phys. Rev. Appl.* 10, 054009 (2018) only investigates the single-excitation transfer in a 4-site 1D chain, which is only an experimental verification of Christandl’s protocol [*Phys. Rev. Lett.* 92, 187902 (2004)]. However, our research aims to address more practical situations, where few-particle quantum states transfer in non-ideal 2D networks with cross-couplings and defects. We not only technically demonstrate a Monte Carlo annealing process to improve the transfer fidelity, but also reveal the underlying physical pictures from the perspectives of quantum chaotic behavior and large-spin representation. Our work stands at the intersection of solid-state quantum processors, state transfer protocols, and quantum many-body physics. As the Reviewer has pointed out, our findings have a good potential to have a considerable impact in these fields.

Following the Reviewer’s suggestion, we explicitly strengthen “what is new” in our research in the last paragraph of the revised introduction.

Comment 1.6: - Role of the inversion symmetry: You say that you use inversion symmetry as a restriction to the optimization algorithm following the idea of previous work performing perfect QST in 1D. But later (e.g. in Supplementary section 8), the inversion symmetry is not utilized. Could you please elaborate on this a bit more deeply? What is its relevance or is it just a way of reducing the space of the parameters? Does inversion symmetry play a central role here? I think it could be interesting to explain this better, i.e. the relationship between inversion symmetry, spatial dimensions, number of excitations, and the presence/absence of defects.

Furthermore, what is it known analytically?

Response 1.6: Inversion symmetry doesn't play a central role in numerical optimizations; it is rather a way of reducing the space of parameters. The Monte Carlo annealing process itself doesn't rely on symmetry. The fewer the constraints, the more freedom to find a good solution, but the cost may be the longer optimization time. For most cases, we use inversion symmetry to reduce the number of parameters, and an optimized fidelity above 0.99 can be easily realized. However, for the small 3×3 network with a defect, only 5 free parameters are left in the presence of inversion symmetry, which makes it challenging to find a good solution. Even if the inversion symmetry condition is removed, we still need a big coupling range ($[-12 \text{ MHz}, -0.3 \text{ MHz}]$) to realize a good state transfer. Indeed, a slightly higher fidelity can be obtained for the Bell state transfer in the 6×6 network if we remove inversion symmetry.

Concerning the usage of inversion symmetry in analytical results, as indicated in Response 1.2, centrosymmetric couplings can be proven to exhibit *perfect* state transfer [see, e.g., *A. Kay, Int. J. Quantum Inform. 8, 641 (2010)*] in 1D systems, as a sufficient and necessary condition. Here with the relaxed condition of *optimal* QST (considering defective couplings, etc.), inversion symmetry is not an encompassing guide to enhance the overall fidelity.

Comment 1.7: Rewriting of Discussion: In the two paragraphs, the first one is lengthy and delves into the theoretical speed limit of the system. As there is no explicit earlier mention throughout the main text of the speed limits, maybe including these details in the Method Section/or and providing some comments in the Result section would work better? Furthermore, the Discussion should in general include some more comments about the results, their relation with previous research, and possible future impact.

Response 1.6: Good suggestion. In the revised manuscript, we have moved the illustration of the quantum speed limit to the 'Methods' section and added more discussions on the relation with previous research and possible future impact.

Minor comments on Introduction:

Comment 1.8: Is "platform-agnostic" of standard use in the quantum technology community? Maybe using platform-independent could be clearer for general readers.

Response 1.8: Okay, "platform-agnostic" has been replaced with "platform-independent" now.

Comment 1.9: Please try to keep the notation for the sites coherent: Eq. (1) starts with "ij" and then you use "n". For example, instead of $J_{i,j}$ and $J_{n,n+1}$, you could simply use $J_{i,j}$ and $J_{i,i+1}$ and additionally denote that $J_{i,i+1}$ refers now to 1D chain.

Response 1.9: Good point. We prefer replacing $\{i, j\}$ with $\{m, n\}$ and have done it in the revised manuscript. To keep notation coherent, we also modified the captions of figures accordingly.

Comment 1.10: The introduction section is quite long, almost 3 pages, covering both general introduction and the theoretical background. It might be useful to restructure it for better readability. Furthermore, it might be useful to have a more concrete text part explaining the structure of the paper. It is now explained quite briefly in the last part of the introduction on page 3.

Response 1.10: We have shortened the introduction by deleting some lengthy descriptions. The last paragraph of the introduction is totally rewritten with a concrete explanation of the structure of our paper.

Comment 1.11: The sentence "establishing a unified and fundamental understanding of perfect QST from angular momentum theory and quantum ergodicity" could be more direct and explicit about the main findings of the manuscript, or you could include a deeper elaboration of it in the discussion section.

Response 1.11: Good suggestion. We have explained this sentence more explicitly by "the optimization of

single-excitation QST yields synchronized precessions among the two mapped spins in the large spin representation” and “the underlying principle governing a perfect QST of few-particle states is immediately connected with the ergodicity breaking by a near Poisson distribution of the ratio of adjacent gaps” in the last paragraph of the introduction.

Minor comments on Results:

Comment 1.12: What does the sentence “the remaining qubits can be switched off at t_{QST} mean exactly? How is it implemented?

Response 1.12: To be clear, we have rephrased this sentence by “the interactions can be switched off at t_{QST} by tuning qubits away from interaction frequency and adjusting NN couplings to the values near zero.”

Comment 1.13: You could add to the sentence “within time scales such that decoherence does not substantially affect device performance” some order of magnitude. Although this information is in the Supplementary material, it is a bit hidden.

Response 1.13: Good point. We have added the relevant time scales explicitly in this sentence by “within time scales ($\sim 0.5 \mu\text{s}$) such that decoherence ($T_1 \approx 140 \mu\text{s}$, $T_2^{\text{SE}} \approx 19 \mu\text{s}$, see Supplementary Section 1) does not substantially affect device performance.”

Comment 1.14: It is not clear what experimental parameters you can manipulate and to what extent. If I understand correctly, you measure the cross-coupling values $J_{m,m'}^x$, which cannot be manipulated. Subsequently, you introduce this value into the optimization algorithm to calculate the NN couplings $J_{i,j}$, which are adjustable. Is my understanding correct? You could explain this in the Method Section more clearly.

Response 1.14: The Reviewer exactly grasps what we do in the manuscript. We explicitly explain this in the revised Method Section “Monte Carlo annealing process”.

Comment 1.15: The references to the Supplementary Sections could include the exact supplementary figures you are referring to.

Response 1.15: We fully agree. This will make our manuscript more readable. We have modified the relevant references following the Reviewer’s suggestion.

Comment 1.16: I do not understand the necessity of introducing the new distance metric for two excitations instead of using Fidelity, as you used in the previous cases. Could you please provide a more detailed explanation? By the way, is this equal or proportional to the so-called Manhattan distance?

Response 1.16: Thanks for the comment. We quantify QST efficiency with fidelity throughout the manuscript, including two-excitation transfer (Fig. 4e of the main text) you mentioned. Note that its value equals to the population probability of the final states for single- and two-excitation cases.

As for the new metric for distance, a generalization of Manhattan distance for two particles (see below), we introduce it for characterizing how few particles (more than one) spread in Fock space because the Manhattan distance, $M[(x_i, y_i), (x_j, y_j)] = |x_j - x_i| + |y_j - y_i|$, is only suitable for single-particle initial states. For two particles, we introduce

$$\begin{aligned} d(|0\rangle, |n'\rangle) &= \frac{1}{l^2} \sum_{i=0}^{l-1} (|x'_i - x_0| + |y'_i - y_0| + |x'_i - x_1| + |y'_i - y_1|) - 1/2 \\ &= \frac{1}{l^2} \sum_{i=0}^{l-1} (M[(x_0, y_0), (x'_i, y'_i)] + M[(x_1, y_1), (x'_i, y'_i)]) - \frac{1}{2}(M[(x_0, y_0), (x_1, y_1)]) \end{aligned}$$

where (x'_i, y'_i) is the cartesian coordinate for each site of the l excitation ($l = 2$) of a generic Fock state $|n'\rangle$. The definition above is an extension of the Manhattan distance for a two-excitation case, which will revert to the

Manhattan distance if $l = 1$. Finally, its main motivation is to provide a quantitative picture of the schematic cartoon figures initially presented in Fig. 1f. For that, one needs to define a space metric in which the fidelity itself is insufficient.

Comment 1.17: What does the term ‘pattern’ mean in the sentence “there is no discernible pattern in the optimized J_{ij} , unlike the case of the 1D protocol.”? Is it not related to mirror (inversion) symmetry? For instance, the circuit depicted in Figure S19 exhibits a certain pattern in this sense. Could you clarify what you mean exactly here?

Response 1.17: Sorry for the misleading sentence. Yes, it is not related to the inversion symmetry. We wrote this sentence in the context of “...it is a ‘black box’, providing no clear physical indication of *why* the optimized coupling solutions give a better QST. Indeed, there is no discernible pattern in the optimized J_{ij} , unlike the case of the 1D protocol. To acquire that insight...” to indicate that we can not find a reason to explain why the optimized $\{J_{ij}\}$ yields a better transfer fidelity by simply looking at the pattern. We have rephrased this sentence in the revised manuscript.

Comment 1.18: Generally: The manuscript uses a notation where operators have hats, then to be consistent, also the density matrix should have a hat in Figures and in the text.

Response 1.18: We modified the figures accordingly.

Response to Reviewer # 2

Comment 2.0: The manuscript seems of high quality, well motivated and written, easily accessible, good plots. Generally, the manuscript thus seem suitable for NAT. Comm.

The results are also of high technical quality: Large number of participating qubits (36), high fidelities of operations, careful optimization of pulse parameters.

Response 2.0: We sincerely thank the Reviewer for evaluating our manuscript being of high quality and recommending the publication in Nature Communications.

Comment 2.1: The concepts and results described in the main text are disconnected from the actual implementation. To improve on this I would move plots such as S11 from the supp mat to the main text.

Response 2.1: Good point. We have added the experimental sequences to the new Fig. 1 of the main text in the revised version.

Comment 2.2: It is good practice to provide information about the quantum device. Very little is said about it. This is important to relate the infidelities to properties of the quantum hardware.

Response 2.2: Thanks for the suggestion. Supplementary Section 1 provides most information about the quantum processor, including energy relaxation time, spin-echo dephasing time, readout fidelity, and idle frequency for each qubit. In the revised version, we add a photograph of our processor, a sketch of the wiring, and a detailed description. We hope it will help the readers better understand our experimental system.

Response to Reviewer # 3

Comment 3.0: The authors previously demonstrated the quantum simulation on the 1D superconducting qubit system [36] based on the theoretical proposal for quantum state transfer on a 1D XY-spin chain [4]. In this paper, they demonstrated the similar experiment on the 2D superconducting qubit system based on the theoretical extension into the multi-dimensional system [40]. The extension from 1D to 2D has implications beyond mere complexity: while 1D systems are integrable, 2D systems with multiple excitations are known to cause chaos, and the required accuracy of the experiments is expected to be high. The authors argue that it is difficult to apply ideal state transfer as in real quantum processors due to the imperfections of the experimental system, and they circumvent this problem by optimizing the coupling strength between qubits using Monte Carlo annealing.

Their system is a square lattice of 6×6 data qubit systems mediated by coupler qubits. Both the data and the coupler qubits are frequency-tunable, effectively allowing the frequency relationship and the coupling strength between the data qubits to be freely manipulated. The lifetime, coherence, readout, and two-qubit gate fidelity in specific qubit pairs for bell-state preparation are described in the appendix 1. In the demonstration, they first prepared the several initial states at the one end of the system, such as single-excitation state, bell state, and two-excitation states. They next transfer these states to the other end of the system within 500 ns. They finally evaluate the state transfer fidelity as 0.90, 0.84, and 0.74, respectively, with the quantum state tomography. In the appendix 3 and 4, they analyze the error source of the state transfer in their experiments. They declare two factors as the possible main error sources: residual thermal excitation after initialization of the qubits (results to $\sim 10\%$ error) and estimation errors in the system parameters (results to tens of % error).

In the discussion, they numerically analyze how the optimized Hamiltonian differs from the Hamiltonian with the random coupling strength. They show the while the Hamiltonian with the random coupling strength show the ergodic feature, the optimized Hamiltonian show the nonergodic feature through their statics of the level spacing distribution. They estimate that their optimization of the coupling strength suppresses chaos and thus increases the state transfer fidelity.

This study has two aspects: quantum simulation of multidimensional XY-spin system and model-based quantum optimal control on the large-scale superconducting qubit system. The proposed method of optimizing state transfer using Monte Carlo annealing is scientifically interesting, but as described in the major revisions below, we are not confident of its practicality. On the other hand, their demonstration shows that they comprehensively understand and control the details of the system, including its tunability and incompleteness. Therefore, we support the acceptance of this manuscript if they address the major revisions listed below.

Response 3.0: We thank the Reviewer for supporting the acceptance of our manuscript and have carefully addressed all of their comments as below.

Major revisions:

Comment 3.1: In the Monte Carlo annealing, they calculated the state transfer infidelity as a cost function by simulating the time evolution of a large-scale quantum system on a classical computer. It clearly does not scale. Is there any prospect to make it scalable?

Response 3.1: We thank the Reviewer for this comment which is precisely in line with Comment 1.4 of Reviewer #1 above. We provide a detailed answer in Response 1.4. In summary, we show that this very simple optimization process can (accompanied by technical improvements) accomplish high-fidelity quantum state transfer in substantially larger system sizes (such as with 1024 qubit lattices for single excitations or 64 qubit lattices for two excitations) as shown in the newly added Section 10 of the Supplementary Materials. As a result, we can easily obtain solutions from the theoretical side for sizes comparable to the largest superconducting circuits to date.

Comment 3.2: Is it possible to visualize the final state of all qubits in the system, not just the initial and final qubits of Fig3 and 4? This information may be helpful to the reader for considering state transfer error sources.

Response 3.2: Good point. Following the suggestion, we plotted the final state of all the qubits for 6×6 single-excitation, 6×6 Bell state, 3×3 two-excitation state and 6×6 two-excitation and added the figure to the revised Supplementary Material.

Comment 3.3: If the state transfer time is extrapolated beyond π , do you see repeated state transfers between the start and end points?

Response 3.3: Yes, there is a back-and-forth of state propagation beyond π (see Fig. 4 for the numerical results). However, its amplitude gradually decays due to imperfections or weak quantum chaotic behaviors.

Comment 3.4: While the proposed method allows state transfer via multiple paths, it does not necessarily result in the fastest state transfer as it does not necessarily employ the maximum coupling achievable in the

system. It is useful to roughly compare the state transfer speed with the case of sequential SWAP (or iSWAP) gates.

Response 3.4: Thanks for the suggestion, which was also pointed out by the Reviewer #1 (see Comment 1.3). Please refer to the Response 1.3 for a detailed comparison between our method, sequential SWAP method, and dynamic circuit method.

Comment 3.5: The author considers only diagonal qubits as NNN coupling, but is there an estimate of the strength of NNN coupling between two adjacent qubits like Q1-Q3?

Response 3.5: Thanks for the comment. NNN couplings between two adjacent qubits like Q1-Q3 are very small, whose absolute values are about 0.1 MHz or less. Numerical simulations suggest that such small couplings tend to slightly decrease the transfer fidelity, typically less than 1%.

Comment 3.6: Have you measured and corrected the flux crosstalk in the system? If crosstalk remains, it may be a source of estimation errors in the qubit frequency and the coupling strength between qubits. Also, please describe how much estimation error the authors expect for the system parameters in appendix 4.

Response 3.6: In our experiments, we have measured and corrected the flux crosstalks from qubits to qubits (≈ 0.0003) and from couplers to qubits (≈ 0.0008), so the errors caused by flux crosstalks mainly come from those uncorrected ones from qubits, couplers to couplers. As illustrated in Supplementary Section 3, the dominant error source is residual thermal population. By subtracting the extrapolated errors caused by thermal populations of 0.5%, the roughly estimated infidelity caused by uncorrected flux crosstalks is around 0.02.

Minor revisions:

Comment 3.7: There is a previous study [A] regarding quantum walks on two-dimensional superconducting qubit system. It would be good to be cited in the main text. [A] M. Gong, et al., Science 372, 948-952 (2021).

Response 3.7: Good suggestion, we have cited this study in the revised introduction of the main text.

Comment 3.8: Please add state transfer execution time to Figure 2.

Response 3.8: Okay, we have added the transfer time in the new Figure 2.

Comment 3.9: Please refer the single qubit gate fidelity of the system in the appendix 1.

Response 3.9: Single-qubit gate fidelity has been added in the Supplementary Section 1.

List of changes

1. Various minor typos have been fixed, and other text modifications prompted by the Reviewers. These are left colorized for easier identification.
2. Inclusion of Section 10 in the Supplementary Materials, addressing the scalability of the optimization approach – it contains Figs. S23, S24 and S25.
3. Figure 1 of the main text includes a new panel (1c) with the experimental sequence, as per the Reviewer's suggestion.
4. Figure 3 has a minor revision on the notation $\rho \rightarrow \hat{\rho}$.
5. The discussion section was rewritten to emphasize the key aspects of our results, leaving the discussion of quantum speed limits to the Methods Section.
6. Supplementary Material now includes Fig. S9 on the measured qubit excited-state population at the QST instant for all cases considered in the main text.
7. Prompted by Reviewer #2 remarks, we included Fig. S1 in the Supplementary Material with details of the experimental device.

Figure 4: Simulated long-time dynamics of QST beyond π . **a**, Population dynamics of Q_1 and Q_{36} for single-excitation QST in the 6×6 quantum network. **b**, Fidelity dynamics of Bell state initially encoded in qubit pairs (Q_1, Q_2) and transferred to (Q_{35}, Q_{36}) in the 6×6 quantum network. **c**, Dynamics of initial qubit pair (Q_1, Q_2) and target qubit pair (Q_{35}, Q_{36}) for the two-excitation QST in the 6×6 quantum network. See Supplementary Section 8 for the couplings for simulating the results in **a**, **b**, and **c**.

REVIEWERS' COMMENTS

Reviewer #1 (Remarks to the Author):

The authors of the manuscript "Enhanced quantum state transfer: Circumventing quantum chaotic behavior" have gone through all the concerns and comments by the reviewers very thoroughly and made changes accordingly. Especially their explanations and comparison tables related to quantum state transfer by constant coupling schemes and by different SWAP gates are very nice.

We recommend publication in Nature Communication.

Reviewer #3 (Remarks to the Author):

The authors have substantially amended the manuscript to incorporate the Referees comments. Their answers have also clarified the remaining points that were ambiguous in the previous version of the manuscript. We recommend publication in Nature communications.

Minor comments:

Table 1 in the response should be added to the paper as it must be useful to the reader.